# TREC: Transient Redundancy Elimination-based Convolution

**Jiawei Guan**◇, **Feng Zhang**◇, **Jiesong Liu**◇, **Hsin-Hsuan Sung**⋆,
**Ruofan Wu**◇, **Xiaoyong Du**◇, **Xipeng Shen**⋆
◇Key Laboratory of Data Engineering and Knowledge Engineering (MOE),
and School of Information, Renmin University of China
⋆Computer Science Department, North Carolina State University
guanjw@ruc.edu.cn, fengzhang@ruc.edu.cn, liujiesong@ruc.edu.cn, hsung2@ncsu.edu,
ruofanwu@ruc.edu.cn, duyong@ruc.edu.cn, xshen5@ncsu.edu

## Abstract

The intensive computations in convolutional neural networks (CNNs) pose challenges for resource-constrained devices; eliminating redundant computations from convolution is essential. This paper gives a principled method to detect and avoid *transient redundancy*, a type of redundancy existing in input data or activation maps and hence changing across inferences. By introducing a new form of convolution (TREC), this new method makes *transient redundancy* detection and avoidance an inherent part of the CNN architecture, and the determination of the best configurations for redundancy elimination part of CNN backward propagation. We provide a rigorous proof of the robustness and convergence of TREC-equipped CNNs. TREC removes over 96% computations and achieves $3.51\times$ average speedups on microcontrollers with minimal (about 0.7%) accuracy loss.

## 1 Introduction

Convolutional Neural Networks (CNNs) are computation intensive, making their deployment on resource-constrained devices (e.g., Microcontrollers equipped with 2MB memory) challenging. Removing computation redundancy in convolutions is hence an important way to speed up CNN inferences.

Depending on where the redundancy comes from, redundancy in CNN can be classified into *lasting redundancy* and *transient redundancy*. Lasting redundancy arises from CNN parameters. As the parameters typically stay unchanged after CNN is deployed for inference, such redundancy can be detected and removed through *offline* methods (e.g., DNN pruning [9, 10, 13, 22, 23, 34, 35] and quantization [15, 17, 26, 33]). Transient redundancy, on the other hand, arises from input data or activation maps, manifesting as similar tiles within an image or an activation map. Unlike *lasting redundancy*, *transient redundancy* has to be detected and removed in *every inference*, elusive to offline methods.

Although *transient redundancy* has been observed in images [25] and videos [16, 36, 37, 38], understanding to it is much less than to *lasting redundancy*. Prior studies have treated *transient redundancy* in an ad-hoc manner. Deep Reuse [25], for instance, diverts, before a convolution, the input images or activation maps from the DNN pipeline to a randomized (to avoid bias) online clustering component to detect the contained redundancy, and then redirects the clustering results back to the DNN. Such ad-hoc treatments work well sometimes on some data, but perform badly in other times (e.g., over 5% accuracy fluctuations in different runs). The reason is the lack of understanding to some fundamental aspects of *transient redundancy*. Examples include

(1) How to reach the optimal setting to minimize the accuracy loss caused by redundancy elimination while maximizing the eliminated redundancy?

(2) How to ensure a stable robust inference performance of *transient redundancy* elimination across runs?

(3) How to minimize the interference imposed by the *transient redundancy* elimination to the result of the converged training of CNN?

This paper proposes a principled way to detect and avoid *transient redundancy* for CNN inference. By introducing a new CNN operator named TREC, it fuses the detection and avoidance of *transient redundancy* into CNN, making them part of its inherent architecture. With the systematic design, finding the best configuration to maximize redundancy-elimination benefits becomes part of the CNN training process, seamlessly integrated as part of the backward propagation. Based on the design, we are able to investigate *transient redundancy* elimination for CNN in a principled way, providing a rigorous proof of the robustness and convergence for TREC, and an intuitive comparative analysis of the computational complexity.

TREC is compatible with both forward and backward propagations, enabling a plug-and-play replacement for convolutional layers in mainstream CNNs without any additional effort. Through a novel LSH back-propagation mechanism, the optimal parameters of TREC are automatically determined; they remain deterministic, clearing the threat to robust performance that hampers practical adoptions of the existing solutions.

We evaluate TREC on diverse popular CNNs, CifarNet [1], SqueezeNet (with and without complex bypass) [14], ZfNet [31], and ResNet-34[12] on a microcontroller (MCU). In particular, ZfNet is preprocessed with channel pruning to meet the strict memory constraints of MCU. Experiments show that by removing 96.25% *transient redundancy*, TREC achieves an average of $4.40\times$ speedup compared to the conventional convolution operator. When applied to the full neural networks, TREC achieves an average of $3.51\times$ speedup with virtually no accuracy loss.

## 2  Related Work

Extensive studies have been conducted to eliminate the *lasting redundancy* of CNNs. Factorization [4, 5, 6, 19, 21, 29, 32], pruning [9, 10, 13, 22, 23, 34, 35], and quantization [15, 17, 26, 33] techniques are some of the important methods. They exploit the redundancy contained in the CNN parameters.

Transient redundancy elimination is complementary to lasting redundancy elimination, dedicated to detecting and removing redundancy from input images or activation maps. It is more challenging to do as it has to happen during online inference. Such redundancy can be further categorized into *temporal* redundancy (e.g., similarities between adjacent video frames) and *spatial* redundancy (e.g., similar tiles within an image). Prior studies [7, 11, 18, 30, 36, 37] have paid attentions mostly to *temporal* similarities between adjacent frames. Deep Reuse [25] is the main method proposed before for detecting and eliminating spatial redundancy within an image for CNN. It takes place as extra operations outside CNN, using Locality-Sensitive Hashing (LSH) with some random hashing vectors for online data clustering. Such an ad-hoc treatment causes important uncertainty about the impact of the clustering errors on the CNN performance. Experimental results show that Deep Reuse causes significant (e.g., 5%) fluctuations on CNN accuracy (detailed in Section 5).

## 3  TREC

In this section, we elaborate on the design of our transient redundancy elimination-based convolution. First, we present the architecture of TREC and how it operates in the forward pass. Then, we delve into the key challenge in the creation of TREC, how to make back-propagation function on TREC-based CNNs.

### 3.1  Basic TREC Architecture

The basic architecture of our proposed TREC is shown in Figure 1. TREC comprises three components in series on top of *Im2col+GEMM* based convolution: LSH-based clustering that reduces the size of the input to the conversion, matrix multiplication of weights and the compressed input, and a recovery step that restores results to the original GEMM output sizes.

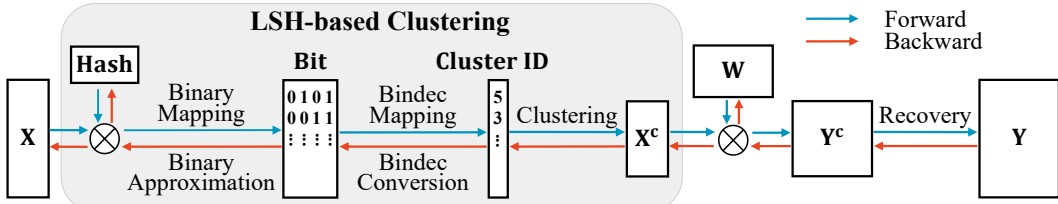

Figure 1: Architecture design of TREC. "$\otimes$" denotes matrix multiplication.

Recall that in *Im2col+GEMM* based convolution, the input feature maps and filters are unfolded to an input matrix $X$ and a weight matrix $W$, and then the convolution is materialized as a matrix multiplication. At a high level, TREC uses Locality-Sensitive Hashing (LSH) to group vectors in the input matrix. It has four steps. First, it applies a hash function matrix *Hash* to $X$. Second, it performs a simple element-wise binary mapping to obtain a bit matrix. Third, each row vector of the bit matrix is converted to a decimal value by bindec mapping (i.e., a mapping from binary to decimal). These integer values indicate the cluster IDs of the neuron vectors in $X$. Fourth, the input matrix $X$ reduces to $X^c$, which is composed of the centroids of clusters. In this way, the size of input matrix is significantly reduced, lowering the computational complexity of the subsequent matrix multiplication. Finally, in the recovery step, vectors in the same cluster reuse the computation results of the corresponding centroids to obtain the final result $Y$.

### 3.2 Breaking the Back-Propagation Barriers for TREC

The key challenges for building a TREC-based CNN are two. The first is in making back-propagation work with TREC. It is challenging because of the discrete nature of the LSH-based clustering in TREC. The second is to understand the impact of TREC on the convergentability of CNN.

We address the first challenge in this section and address the second challenge in Section 4. At a high level, our strategy consists of two key components. First, we replace the piecewise mappings in TREC with similar-shaped functions as binary approximation approaches. Second, we propose a bindec conversion algorithm to resolve the complexities in obtaining centroids. Before explaining them, we first give a closer examination of the operations in the forward pass of the basic TREC, which will help the follow-up explanations.

**A Closer Look at Redundancy Elimination in the Forward Pass.** Figure 2 illustrates the redundancy elimination in the forward pass of the basic TREC. Specifically, let $X \in \mathbb{R}^{4 \times 6}$ be an unfolded input matrix after *Im2col*. TREC first slices $X$ into $L$ groups vertically ($L = 2$ in Figure 2 *Left*), so each sub-matrix is of size $4 \times 3$. The slicing reduces vector length which can often allow the clustering to identify more similar vectors [24]. Applying the LSH clustering to each sub-matrix produces the centroid matrix $X^c$.

LSH is chosen because as a lightweight online clustering method, it produces good clustering results without incurring excessive overhead. Figure 2 *Right* shows more details of LSH clustering with an example. $X_1$ is the first sub-matrix of $X$ in Figure 2 *Left*. For each input vector $x$ in $X_1$, it is hashed through the following function parameterized with a random vector $v$:

$$h_v(x) = \begin{cases} 1 & if \quad v \cdot x > 0 \\ 0 & if \quad v \cdot x \leq 0 \end{cases} \tag{3.1}$$

First, $X_1$ is transformed into a projected matrix with $H = 2$ random hash vectors (i.e., 2 column vectors of the hash matrix). Second, the projected vectors are mapped to bit vectors with $2^H$ possibilities, as defined by Eq. 3.1. Third, the bindec rule converts each bit vector to an integer value, which can be used as the cluster ID of vector $x$. In this way, similar vectors have a high chance of being mapped to the same cluster. Finally, each vector in $X^c$ is computed using vectors with the same cluster ID. Note that in practice, $H$ is much smaller than the length of neural vectors, so the overhead introduced by LSH is low.

**Binary Approximation.** To solve the problem of discontinuity in binary mapping during back-propagation, we introduce binary approximation to make it differentiable. The binary mapping handles the projected matrix in an element-wise manner, which acts as a binary classification that sets

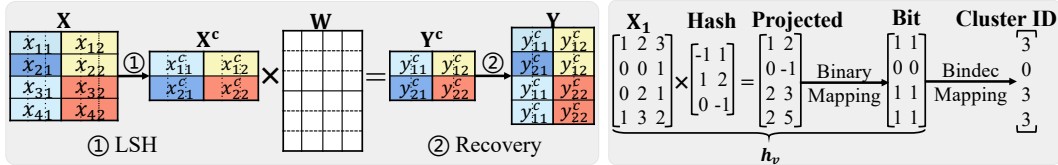

Figure 2: *Left*: The illustration of redundancy elimination. In matrix $X$, vectors in the same color belong to the same cluster. *Right*: An illustration of LSH. $X_1$ is the first sub-matrix of $X$ on the left.

elements greater than zero to one, and otherwise to zero. To that end, we employ a sigmoid function as a substitute of the binary mapping:

$$S(x) = \frac{1}{1 + e^{-a \times (x-b)}}, \tag{3.2}$$

where $a$ denotes the slope and $b$ denotes the abscissa value of the central point of the curve. The sigmoid function suits TREC because it is 1) monotonic and bounded; 2) smooth and differentiable. Meanwhile, the sigmoid function offers a parameterized way to materialize the effects of binary mapping for TREC. The parameters $a$ and $b$ in the sigmoid function are hyperparameters determined empirically. When setting them, the rationale worth mentioning is that the value of $a$ shall be large such that as few points as possible appear at the mid-slope of the sigmoid function, which would allow positive and negative numbers to approach one and zero respectively. And the value of $b$ shall shift the function slightly to the right, which would allow the value at $x = 0$ to approach zero. The bit vectors can be then approximated by an application of such a sigmoid function to the projected matrix.

**Bindec Conversion.** Another step that impedes back-propagation for TREC is bindec mapping, that is, the rightmost step in the example in Figure 2, which converts every binary vector into its corresponding integer value. To make this process differentiable, we design a transformation matrix as follows:

$$\begin{bmatrix} 2^{H-1}/1 & 2^{H-1}/2 & \cdots & 2^{H-1}/2^H \\ \vdots & \vdots & \ddots & \vdots \\ 2^1/1 & 2^1/2 & \cdots & 2^1/2^H \\ 2^0/1 & 2^0/2 & \cdots & 2^0/2^H \end{bmatrix} \in \mathbb{R}^{H \times 2^H}. \tag{3.3}$$

Eq. 3.3 shows that elements in the same row of the transformation matrix have the same numerator, and elements in the same column have the same denominator. Horizontally, the denominators increase from 1 to $2^H$. Vertically, the numerators are $[2^{H-1}, ..., 2^1, 2^0]$ from top to bottom, which are the coefficients when converting a binary number to decimal.

Figure 3: The overall idea of constructing the $Avg$ matrix.

The overall idea of bindec conversion is shown in Figure 3, which follows the example of Figure 2. First, we add 1 to the rightmost column of the bit matrix. In this way, the range of the integers changes from $[0, 2^H - 1]$ to $[1, 2^H]$, which is consistent with the denominators of the transformation matrix. The result matrix is then multiplied with the transformation matrix, which can be seen as two steps: 1) multiply the bit vectors with the numerators of the transformation matrix's column vectors to obtain integer values; 2) divide the integer values by the denominators of the column vectors. As a

result, in the quotient matrix, for row $i$ where vector $x_i$ is located, $Quotient_{i,j} = 1$ appears only at the position where the column number $j$ equals $x_i$'s cluster ID. For example, in Figure 3, the bit vector in the second row (i.e., row 1) is $[0, 0]$, and the cluster ID is 0. In row 1 of the quotient matrix, only column 0 has 1, i.e., $Quotient_{1,0} = 1$.

The method can successfully do the mapping. But as $H$ increases, the dimension of the transformation matrix increases exponentially, resulting in a huge memory waste. To ease the space and computation, we apply the above process only in the backward pass, while the forward pass still uses binary and bindec mappings. This means that the results of the forward pass such as cluster IDs, the number of vectors per cluster (denoted as $N_c$), and the number of clusters (denoted as $C$), can be reused by back-propagation. Therefore, we can directly construct a transformation matrix of size $H \times C$, where each column represents a known cluster ID that is derived from the forward calculation. As shown in the lower part of Figure 3, the use of forward information can significantly avoid space waste.

Now, we can obtain an index matrix to indicate the vector-cluster relationship. We can do that by applying a element-wise function to the quotient matrix, which gives 1 if the element is 1 and 0 otherwise. Such a function is however not differentiable. To solve this problem, we introduce an impulse-like Gaussian function of the following form:

$$G(x) = e^{-\frac{(x-1)^2}{2\sigma^2}}, \tag{3.4}$$

It is a symmetrical bell-shaped curve centered at $x = 1$, with $\sigma$ controlling its width. With a small $\sigma$, the Gaussian function can be used to approximate the desired index matrix. Then, we divide the index matrix by the number of vectors per cluster $N_c$, and we obtain an average matrix denoted as $Avg$. Finally, the cluster centroids can be calculated as follows:

$$X^c = Avg^T \cdot X. \tag{3.5}$$

For the example in Figure 2, we get the cluster centroids as follows:

$$\begin{bmatrix} \frac{1}{3} & 0 & \frac{1}{3} & \frac{1}{3} \\ 0 & 1 & 0 & 0 \end{bmatrix} \cdot \begin{bmatrix} 1 & 2 & 3 \\ 0 & 0 & 1 \\ 0 & 2 & 1 \\ 1 & 3 & 2 \end{bmatrix} = \begin{bmatrix} \frac{2}{3} & \frac{7}{3} & 2 \\ 0 & 0 & 1 \end{bmatrix} \tag{3.6}$$

**Parameter Setting.** TREC uses Sigmoid and Gaussian functions to simulate the mappings. As mentioned, TREC needs the range of function parameters to meet certain conditions, such as using a small $\sigma$ to narrow the width of the Gaussian function. However, fixing a $\sigma$ value is subject to gradient explosion, resulting in a large gradient variance and convergence difficulties. Therefore, we propose a solution that dynamically sets $\sigma$ based on $H$.

We use Figure 4 to illustrate the gradient explosion problem. The curves are of a Gaussian function with $\sigma = 1$ and its derivative. We can see that the derivative function varies widely on both sides of $x = 1$, and its maximum and minimum values occur at $x = 1 \pm \sigma$. The value range of the derivative function increases as $\sigma$ increases. Therefore, manually setting a fixed $\sigma$ has a high chance of incurring a large gradient variance.

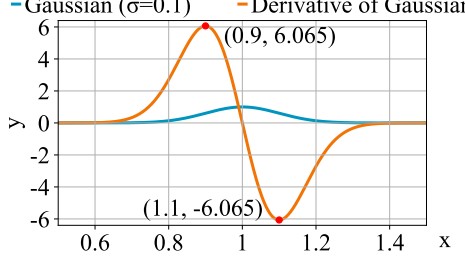

Figure 4: Gaussian function and its derivative with $\sigma = 1$.

To address this problem, we first analyze the quotient matrix. According to the rules of the transformation matrix, the quotient matrix takes values in the range $[\frac{1}{2^H}, 2^H]$, and the difference is at least $\frac{1}{2^H}$. Moreover, the $i^{th}$ vector has $Quotient_{i,j} = 1$ when the cluster ID is equal to column $j$. Based on the observations, we avoid producing large gradients simply by ensuring

$$\sigma < \frac{1}{k \cdot 2^H} \quad and \quad k > 1. \tag{3.7}$$

Consequently, the Gaussian function not only provides an ideal approximation of the index matrix, but also ensures the stability of the training under any $H$ value.

# 4 Theoretical Analysis

In this section, we provide a theoretical analysis of TREC in terms of robustness, convergence, and complexity. All proofs are available in the Appendix.

## 4.1 Robustness

Neural networks are sensitive to inputs [8, 27]. In practice, a small perturbation of the input image can misdirect the neural network and significantly decrease its classification accuracy. To prove that TREC does not impair the stability of the convolutional layer, we use the *Lipschitz constant*, which is commonly used to assess the robustness of neural networks, to upper bound the relationship between the input perturbation and output variation [28]. In the following, we use $\|X\|_\infty$ to denote the $L_\infty$-norm of the matrix space $\mathbb{R}^{m \times n}$.

**Definition 1.** *A function $f \colon \mathbb{R}^{m_1 \times n_1} \to \mathbb{R}^{m_2 \times n_2}$ is called Lipschitz continuous if there exists a constant $L > 0$ such that*

$$\forall X, Y \in \mathbb{R}^{m \times n}, \quad \|f(X) - f(Y)\|_\infty \leq L\|X - Y\|_\infty. \tag{4.1}$$

The smallest $L$ that makes the previous inequality true is called the *Lipschitz constant* of $f$ and can be denoted as $L(f)$.

Suppose $Conv \colon X \to X \cdot W$ is a conventional convolution operator, and $TREC \colon X \to Avg \cdot X \cdot W$ is our proposed method. Recall the description of the $Avg$ matrix in Section 3.2 and Eq. 3.6; $\sum_i Avg_{i,j} \leq 1$ is satisfied for any column $j$. Combining Definition 1 with the above discussion, we have:

$$L(Conv) = \|W\|_\infty, \quad L(TREC) = \|W \otimes Avg^T\|_\infty \leq \|W\|_\infty, \tag{4.2}$$

where $\otimes$ denotes the Kronecker product. In summary, under the same network configuration, the Lipschitz constant of TREC is no larger than that of the conventional convolution. That is to say, TREC has the ability to maintain the robustness of convolutional layers.

## 4.2 Convergence

In this section, we provide a theoretical analysis of the convergence of TREC-equipped CNNs. Analyzing the convergence of the above CNNs directly is difficult, since the specific CNN architectures are unknown except for the convolutional layers. Instead, we address this problem by building a connection with stochastic gradient descent, and prove that the proposed method is guaranteed to converge under reasonable assumptions. In the following, we use $\|x\|_2$ to denote the $L_2$-norm of the Hilbert space $\mathbb{R}^n$.

We assume a CNN network with $R$ TREC layers and $T$ other layers. Without loss of generality, we use $W = [W_1, \ldots, W_{R+T}] \in \mathbb{R}^{d_1}$ to denote the weights of all layers, and $H = [H_1, \ldots, H_R] \in \mathbb{R}^{d_2}$ to denote the hash functions of all convolutional layers. In accordance with the method [2], we represent a sample (or set of samples) as a random seed $\xi$, and refer to the loss for a given $(W, H, \xi)$ as $f(W, H; \xi)$. In addition, given a dataset with $n$ samples, we refer to the objective function $F : \mathbb{R}^D \to \mathbb{R}$ as either $F(W, H) = \mathbb{E}[f(W, H; \xi)]$ or $F(W, H) = \frac{1}{n} \sum_{i=1}^n f_i(W, H)$.

Following the stochastic gradient descent, we define the parameter update rule at iteration $k$ as:

$$W_{k+1} \leftarrow W_k - \alpha_k \cdot g(W_k, H_k; \xi_k), \quad H_{k+1} \leftarrow H_k - \alpha_k \cdot q(W_k, H_k; \xi_k), \tag{4.3}$$

where $\alpha_k$ denotes the stepsize, $g(W_k, H_k; \xi_k)$ and $q(W_k, H_k; \xi_k)$ represent the gradients of the loss function with respect to $W_k$ and $H_k$ separately, which can be the stochastic gradients or the mean gradients of a mini-batch:

$$g(W_k, H_k; \xi_k) = \begin{cases} \nabla f_W(W_k, H_k; \xi_k) \\ \frac{1}{n_k} \sum_{i=1}^{n_k} \nabla f_W(W_k, H_k; \xi_k) \end{cases}, q(W_k, H_k; \xi_k) = \begin{cases} \nabla f_H(W_k, H_k; \xi_k) \\ \frac{1}{n_k} \sum_{i=1}^{n_k} \nabla f_H(W_k, H_k; \xi_k). \end{cases} \tag{4.4}$$

Now we summarize the key assumptions required to establish our results.

**Assumption 1** (Lipschitz-continuous objective gradients)**.** *The objective function $F : \mathbb{R}^D \to \mathbb{R}$ is continuously differentiable and the partial derivatives of $F$, namely, $\nabla F_W : \mathbb{R}^{d_1} \to \mathbb{R}^{d_1}$ and $\nabla F_H : \mathbb{R}^{d_2} \to \mathbb{R}^{d_2}$, are Lipschitz continous with Lipschitz constants $L > 0$, i.e.,*

$$\|\nabla F_W(W, H) - \nabla F_W(\overline{W}, H)\|_2 \leq L\|W - \overline{W}\|_2, \quad \text{for all } \{W, \overline{W}\} \subset \mathbb{R}^{d_1},$$
$$\|\nabla F_H(W, H) - \nabla F_H(W, \overline{H})\|_2 \leq L\|H - \overline{H}\|_2, \quad \text{for all } \{H, \overline{H}\} \subset \mathbb{R}^{d_2}. \tag{4.5}$$

Assumption 1 ensures that the partial derivatives of $F$ do not change arbitrarily fast with respect to the parameters, thus well indicating how far to move to decrease $F$. To move forward, we denote the variances of $g(W_k, H_k; \xi_k)$ and $q(W_k, H_k; \xi_k)$ as:

$$\mathbb{V}_{\xi_k}[h(W_k, H_k; \xi_k)] := \mathbb{E}_{\xi_k}\left[\|h(W_k, H_k; \xi_k)\|_2^2\right] - \left\|\mathbb{E}_{\xi_k}[h(W_k, H_k; \xi_k)]\right\|_2^2, \quad h \in \{g, q\}. \qquad (4.6)$$

**Assumption 2** (First and second moment limits). *The objective function and the stochastic gradient algorithm satisfy the following conditions:*

(a) *The sequence of iterations $\{(W_k, H_k)\}$ is contained in an open set over which $F$ is bounded below by a scalar $F_{inf}$.*

(b) *There exist scalars $\mu_G \geq \mu > 0$ such that, for all $k \in \mathbb{N}$,*

$$\nabla F_W(W_k, H_{k+1})^T \mathbb{E}_{\xi_k}[g(W_k, H_k; \xi_k)] \geq \mu\|\nabla F_W(W_k, H_k)\|_2^2, \qquad (4.7a)$$

$$\nabla F_H(W_k, H_k)^T \mathbb{E}_{\xi_k}[q(W_k, H_k; \xi_k)] \geq \mu\|\nabla F_H(W_k, H_k)\|_2^2, \qquad (4.7b)$$

$$\|\mathbb{E}_{\xi_k}[g(W_k, H_k; \xi_k)]\|_2 \leq \mu_G\|\nabla F_W(W_k, H_k)\|_2, \qquad (4.7c)$$

$$\|\mathbb{E}_{\xi_k}[q(W_k, H_k; \xi_k)]\|_2 \leq \mu_G\|\nabla F_H(W_k, H_k)\|_2. \qquad (4.7d)$$

(c) *There exist scalars $M \geq 0$ and $M_V \geq 0$ such that, for all $k \in \mathbb{N}$,*

$$\begin{aligned}\mathbb{V}_{\xi_k}[g(W_k, H_k; \xi_k)] \leq M + M_V\|\nabla F_W(W_k, H_k)\|_2^2, \\ \mathbb{V}_{\xi_k}[q(W_k, H_k; \xi_k)] \leq M + M_V\|\nabla F_H(W_k, H_k)\|_2^2.\end{aligned} \qquad (4.8)$$

We first analyze the convergence with a fixed stepsize and prove that the CNNs equipped with TREC can converge sub-linearly to the neighborhood of the critical points for the non-convex problem under the above assumptions.

**Theorem 1** (Fixed stepsize). *Assume that Assumptions 1 and 2 hold, and the fixed stepsize is $\alpha_k = \bar{\alpha}$ for all $k \in \mathbb{N}$ satisfying $0 < \bar{\alpha} \leq \dfrac{\mu}{LM_G}$. Then, the average-squared partial derivatives of $F$ corresponding to the stochastic gradient iterations satisfy the following inequality for all $K \in \mathbb{N}$:*

$$\frac{1}{K}\sum_{k=1}^{K}\mathbb{E}\left[\|\nabla F_W(W_k, H_k)\|_2^2 + \|\nabla F_H(W_k, H_k)\|_2^2\right] \leq \frac{2\bar{\alpha}LM}{\mu} + \frac{2(F(W_1, H_1) - F_{inf})}{K\mu\bar{\alpha}}. \qquad (4.9)$$

Therefore, the optimal solution we can obtain is dominated by $\frac{2\bar{\alpha}LM}{\mu}$. Further, we also prove the convergence with a dimishing stepsize that meets the requirements in the study [2]:

$$\lim_{K \to \infty}\sum_{k=1}^{K}\alpha_k = \infty \quad and \quad \lim_{K \to \infty}\sum_{k=1}^{K}\alpha_k^2 < \infty. \qquad (4.10)$$

**Theorem 2** (Dimishing stepsize). *Assume that Assumptions 1 and 2 hold and the dimishing stepsize sequence $\{\alpha_k\}$ satisfies Eq. 4.10. Then, with $A_K := \sum_{k=1}^{K}\alpha_k$,*

$$\mathbb{E}\left[\frac{1}{A_K}\sum_{k=1}^{K}\alpha_k\left(\|\nabla F_W(W_k, H_k)\|_2^2 + \|\nabla F_H(W_k, H_k)\|_2^2\right)\right] \leq \frac{2(F(W_1, H_1) - F_{inf})}{\mu A_k} + \frac{2LM}{\mu A_k}\sum_{k=1}^{K}\alpha_k^2 \xrightarrow{K \to \infty} 0. \qquad (4.11)$$

**Remark 1.** *Theorem 2 has shown the decaying of gradients $\|\nabla F_W(W_k, H_k)\|_2$ and $\|\nabla F_H(W_k, H_k)\|_2$ even with noise when the stepsize is dimishing. This is because the second condition in Eq. 4.10 implies the right-hand side of Eq. 4.11 to converge to a finite limit as $K$ increases.*

Theorems 1 and 2 prove convergence in the limit. Taking a step forward, we study the theoretical bounds to understand the convergence of TREC-equipped CNNs in practical applications. According to Eq. 4.9 and 4.11, as long as the practical application satisfies Assumptions 1 and 2, we can obtain a theoretical upper bound of the sum of squared gradients without considering the limit. The relevance of this theoretical upper bound to practical applications is reflected in the values of the constants such as $L$, $\mu$, $M$, and the specific setting of the learning rate. From Eqs. 4.5- 4.8, the constants $L$, $M$, and $M_G = M_v + \mu_G^2$, which are greater than 0, exist according to the L2-norm. Through experimental verification, we find that the constant $\mu$ is always greater than 0 during the training process and gradually converges to 1 with time, satisfying the sufficient direction assumption. We present in Appendix B the trend of the constant $\mu$.

## 4.3 Complexity

We show in Table 1 an intuitive comparative analysis of two major complexity metrics. The complexity of binary and bindec mappings are not taken into account because they are parameterless and insignificant compared with other computational overheads. For the parameter count, TREC exceeds the conventional convolution by a hash matrix of $O(L \cdot H)$ size, with trivial KB-level space occupancy. For the computation complexity, we define *redundancy ratio*: $r_t = 1 - N_c/N$ as the size reduction of the input matrix to indicate the fraction of *transient redundancy* within input images or activation maps. As long as $H < M \cdot r_t$ is satisfied, TREC can yield computational benefits. In practical validation (Section 5.2), we notice that the reduction of *transient redundancy* of a single convolutional layer reaches up to 96.25%, which brings confidence to the advantages of TREC.

Table 1: Parameter counts and FLOPs for conventional GEMM-based convolution and TREC. The input and weight matrices unfolded by Im2col are of size $N \times K$ and $K \times M$. The hash matrix of TREC is of size $L \times H$. $N_c$ denotes the number of clusters.

|  | Parameters | FLOPs |
|---|---|---|
| Conventional | $K \cdot M$ | $N \cdot K \cdot M$ |
| TREC | $K \cdot M + L \cdot H$ | $(\frac{H}{M} + 1 - r_t) \cdot N \cdot K \cdot M$ |

# 5 Evaluation

## 5.1 Experimental Setup

**Methodology.** To demonstrate the efficacy of TREC on CNN inference acceleration, we first apply TREC to single convolutional layers to measure single-layer speedups and accuracy. Second, we apply TREC to the complete neural networks to measure end-to-end inference performance. Third, we analyze the influence of different factors on performance, including the dynamic setting of the Gaussian parameter, and the clustering configurations.

**Platforms.** As computation reduction is most needed on resource constrained devices, we focus our evaluation on Microcontrollers (MCU). Specifically, all inferences are performed on an STM32F469NI MCU with 324KB SRAM and 2MB Flash, using the CMSIS-NN kernel optimized for Arm Cortex-M devices [20]. All trainings are performed using PyTorch 1.10.1 (open-source software with a BSD license) on a machine equipped with a 20-core 3.60GHz Intel Core i7-12700K processor, 128GB of RAM, and an NVIDIA GeForce RTX A6000 GPU with 48 GB memory.

**Workloads.** We evaluate TREC with five compact CNNs that fit MCU, CifarNet [1], ZfNet [31], SqueezeNet with and without complex bypass [14], and ResNet-34 [12]. Most trainings and inferences are performed on CIFAR-10, while for ResNet, we use the downsampled ImageNet [3] with 64×64 resolution (since the original large images cause ResNet-34 to run out of MCU memory). All networks are optimized by SGD. The learning rate starts from 0.001 and decreases by 0.1 every 15 epochs. The batch size, weight decay, and momentum are set to 256, 0.9, and $10^{-4}$, respectively, and the maximal epoch is set to 100. The batch size at inference is 1 due to the stringent memory limit on MCU. It is worth noting that ZfNet is preprocessed with channel pruning to meet the strict memory constraints of MCU. It also allows us to see how well the transient redundancy elimination (TREC) works in the presence of lasting redundancy elimination (pruning).

## 5.2 Single-Layer Performance

We report in Appendix C the single-layer speedups and accuracy loss achieved by replacing the conventional convolution with TREC. For comparison, we also include the results from Deep Reuse [25], a state-of-the-art method in avoiding *transient redundancy* in CNN. We have the following observations. First, TREC detects and eliminates about 96.25% of the *transient redundancy*. Second, TREC brings significant performance benefits for individual layers with an average speedup of 4.40×. Especially for the SqueezeNet expand layers, TREC gets up to 18.66× speedup, indicating that TREC is especially beneficial to computation-intensive layers. The speedup is lower than the ratio of *transient redundancy* that TREC eliminates, due to the overhead introduced by the clustering. Third, the influence on accuracy by TREC is small and mixed. When applied to some layers, it

may cause around 0.8% accuracy drop, but it may also give slight accuracy improvements when applied to some other layers. TREC overall preserves the accuracy much better (about 3.33% on average) than Deep Reuse does. Furthermore, the *Conf.* columns report the LSH configurations in our experiments (Section 5.4 gives more details on the impacts of the hyperparameters $L$ and $H$). We also illustrate the impact of batch size on performance by measuring the redundancy ratio of a single layer in Appendix C.2 , which shows that TREC has strong redundancy removal ability at any batch size, and that this ability increases with batch size. All above results demonstrate that TREC can exert its full potential, achieving significant speedups while minimizing the accuracy loss.

## 5.3 End-to-End Performance

Table 2 compares the end-to-end performance brought by TREC, Deep Reuse, and conventional convolution. Given the inherent randomness of Deep Reuse, we present its result interval of 150 runs. Note that after channel pruning, the baseline accuracy of ZfNet is slightly lower than its standard accuracy of 83%, but the model size is 7.6% of the original model, which can well meet the memory constraint of MCU.

Table 2: End-to-end performance comparison.

| Network | Conv Method | Average Time per Image (ms) | Top-1 Accuracy (%) |
|---|---|---|---|
| CifarNet | Conventional | 217.32 | 78.2 |
| | Deep Reuse | 154.44 | $73.2 \sim 76.1$ |
| | TREC | 153.92 | 76.5 |
| ZfNet | Conventional | 3557.32 | 80.1 |
| | Deep Reuse | 814.03 | $72.5 \sim 76.6$ |
| | TREC | 814.01 | 78.9 |
| Vanilla SqueezeNet | Conventional | 1639.51 | 83.5 |
| | Deep Reuse | 328.97 | $79.8 \sim 81.9$ |
| | TREC | 327.90 | 83.0 |
| SqueezeNet + Complex Bypass | Conventional | 1998.86 | 85.3 |
| | Deep Reuse | 543.71 | $80.5 \sim 83.1$ |
| | TREC | 544.03 | 84.6 |
| ResNet-34 (ImageNet-64×64) | Conventional | 4242.77 | 52.6 |
| | Deep Reuse | 1379.26 | $46.7 \sim 49.9$ |
| | TREC | 1378.75 | 52.2 |

Several findings can be drawn from Table 2. First, compared to the conventional convolution, TREC achieves 3.51× speedup on average with minor (about 0.7%) accuracy loss. Second, TREC and Deep Reuse have comparable inference time, but TREC is superior in terms of stable performance and higher accuracy. Third, TREC cuts the accuracy loss of Deep Reuse from as much as 7.6% to 0.4~1.7%, an up to 84% reduction.

## 5.4 Configuration Analysis

**Dynamic and Static $\sigma$ of Gaussian.** In this set of experiments, we measure the effect of the strategy proposed in Section 3.2 to dynamically set $\sigma$ based on $H$. To eliminate the influence of irrelevant factors, we choose the most computation-intensive convolutional layers among four networks and replace them with TREC. As shown in Figure 5, we compare our proposed dynamic strategy with the static approach that manually sets a fixed $\sigma$ value of $10^{-4}$. As seen, the accuracy of the models using a fixed $\sigma$ drops sharply after $H = 12$, and it shows a decreasing trend as $H$ increases. Recalling the discussion in Section 3.2, when $H > 12$, we have $\sigma = 10^{-4} > \frac{1}{k \cdot 2^H}$. Therefore, the gradient of the hash matrix tends to fall into the maximum value range of the Gaussian derivatives, resulting in a large gradient variance and difficulty in model convergence. Thus, our dynamic $\sigma$ strategy can effectively guarantee the performance.

**Impacts of Hyperparameters $L$ and $H$** Two hyperparameters in TREC are 1) $L$: the width of the sub-matrices obtained by slicing, and 2) $H$: the number of hash functions. In application of TREC, they can be chosen empirically like other CNN hyperparameters. In this part, we provide some observations on their impacts. Figure 6 shows the accuracy of TREC, Deep Reuse, and conventional convolution under different $L$ and $H$ configurations. The upper part is the result of changing $H$ with

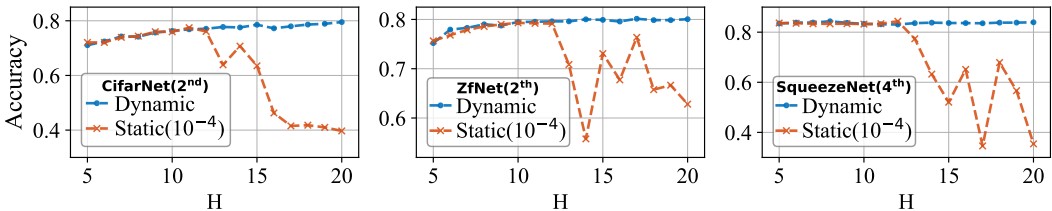

Figure 5: Effects of dynamic and static settings of $\sigma$.

fixed $L$, and the range of $H$ is set to 5 to 20. The lower part is the result of changing $L$ with fixed $H$, and the range of $L$ changes according to the input and weight dimensions of different layers. We have the following observations: 1) TREC and Deep Reuse show a consistent trend of accuracy, which goes up as $H$ increases, and goes down as $L$ increases. 2) TREC has robust and stable inference performance, which is preferable to the non-deterministic Deep Reuse. 3) TREC is more inclined to stay at high accuracy, proving its ability to mine richer data features.

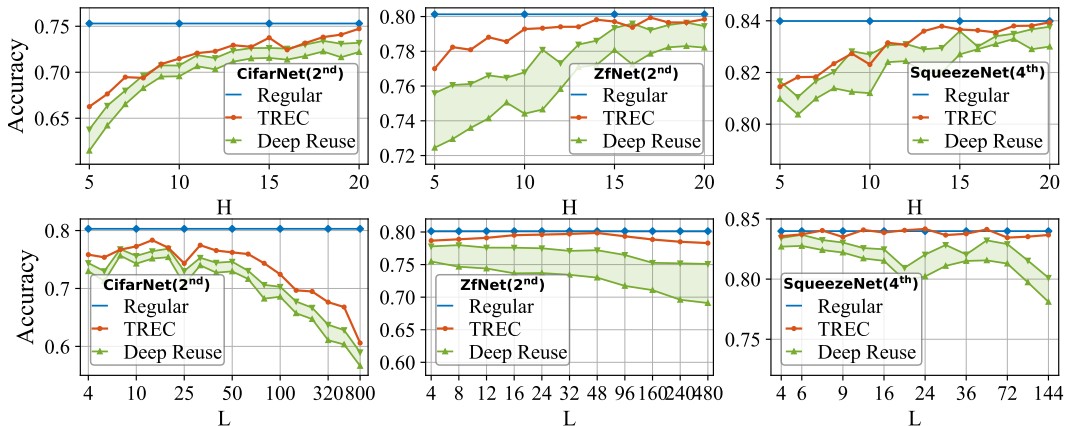

Figure 6: Influence of LSH parameter configuration on accuracy.

## 6  Conclusion

In this paper, we propose a principled approach to detecting and avoiding *transient redundancy*. It introduces a new form of convolutional operator called TREC, which integrates the redundancy avoidance as well as the optimal configuration determination into CNN as an intrinsic part of the CNN architecture. We experimentally and theoretically demonstrate the effectiveness of TREC. This new approach may open many new possibilities for CNN on resource-constrained devices. As a foundational research outcome, it is neutral in terms of societal impacts. A limitation of this study is that it focuses on CNNs, but transient redundancy may also exist in other types of Neural Networks as well, exploration of which is left for future to explore.

## Acknowledgments and Disclosure of Funding

This material is supported by the National Natural Science Foundation of China (No. 62072458, 62172419, 61732014, and 62072459) and Beijing Nova Program, and is based upon work supported by the National Science Foundation (NSF) under Grants CCF-1703487, CCF-2028850 and CNS-1717425. Any opinions, findings, and conclusions or recommendations expressed in this material are those of the authors and do not necessarily reflect the views of NSF. Feng Zhang is the corresponding author of this paper.

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
