# A Proofs

**Lemma 1.** *Assume that Assumptions 1 and 2 hold, the iterations satisfy the following inequality for all $k \in \mathbb{N}$:*

$$\mathbb{E}_{\xi_k}[F(W_{k+1}, H_{k+1})] - F(W_k, H_k) \leq -(\mu - \frac{1}{2}\alpha_k L M_G)\alpha_k [\|\nabla F_W(W_k, H_k)\|_2^2 + \|\nabla F_H(W_k, H_k)\|_2^2]$$
$$+ \alpha_k^2 L M.$$

(A.1)

*Proof.* Under Assumption 1, for any $\{W, \widetilde{W}\} \in \mathbb{R}_1^d$ and $\{H, \widetilde{H}\} \in \mathbb{R}_2^d$, one obtains

$$F(W, H) - F(\widetilde{W}, \widetilde{H}) = (F(W, H) - F(\widetilde{W}, H)) + (F(\widetilde{W}, H) - F(\widetilde{W}, \widetilde{H}))$$

$$= \int_0^1 \frac{\partial F(\widetilde{W} + t(W - \widetilde{W}), H)}{\partial t} dt + \int_0^1 \frac{\partial F(\widetilde{W}, \widetilde{H} + t(H - \widetilde{H}))}{\partial t} dt$$

$$= \int_0^1 \nabla F_W(\widetilde{W} + t(W - \widetilde{W}), H)^T (W - \widetilde{W}) dt + \int_0^1 \nabla F_H(\widetilde{W}, \widetilde{H} + t(H - \widetilde{H}))^T (H - \widetilde{H}) dt$$

$$= \nabla F_W(\widetilde{W}, H)^T (W - \widetilde{W}) + \int_0^1 [\nabla F_W(\widetilde{W} + t(W - \widetilde{W}), H) - \nabla F_W(\widetilde{W}, H)]^T (W - \widetilde{W}) dt$$

$$+ \nabla F_H(\widetilde{W}, \widetilde{H})^T (H - \widetilde{H}) + \int_0^1 [\nabla F_H(\widetilde{W}, \widetilde{H} + t(H - \widetilde{H})) - \nabla F_H(\widetilde{W}, \widetilde{H})]^T (H - \widetilde{H}) dt$$

$$\leq \nabla F_W(\widetilde{W}, H)^T (W - \widetilde{W}) + \int_0^1 L\|t(W - \widetilde{W})\|_2 \|W - \widetilde{W}\|_2 dt$$

$$+ \nabla F_H(\widetilde{W}, \widetilde{H})^T (H - \widetilde{H}) + \int_0^1 L\|t(H - \widetilde{H})\|_2 \|H - \widetilde{H}\|_2 dt$$

$$\leq \nabla F_W(\widetilde{W}, H)^T (W - \widetilde{W}) + \nabla F_H(\widetilde{W}, \widetilde{H})^T (H - \widetilde{H}) + \frac{1}{2}L(\|W - \widetilde{W}\|_2^2 + \|H - \widetilde{H}\|_2^2).$$

Therefore, the iterations generated by stochastic gradient algorithm satisfy

$$F(W_{k+1}, H_{k+1}) - F(W_k, H_k) \leq \nabla F_W(W_k, H_{k+1})^T (W_{k+1} - W_k) + \frac{1}{2}L\|W_{k+1} - W_k\|_2^2$$

$$+ \nabla F_H(W_k, H_k)^T (H_{k+1} - H_k) + \frac{1}{2}L\|H_{k+1} - H_k\|_2^2$$

$$\leq -\alpha_k \nabla F_W(W_k, H_{k+1})^T g(W_k, H_k; \xi_k) + \frac{1}{2}\alpha_k^2 L\|g(W_k, H_k; \xi_k)\|_2^2$$

$$- \alpha_k \nabla F_H(W_k, H_k)^T q(W_k, H_k; \xi_k) + \frac{1}{2}\alpha_k^2 L\|q(W_k, H_k; \xi_k)\|_2^2.$$

Taking expectations in these inequalities with respect to the distribution of $\xi_k$, and noting that $(W_{k+1}, H_{k+1})$—but not $(W_k, H_k)$—depends on $\xi_k$, we obtain

$$\mathbb{E}_{\xi_k}[F(W_{k+1}, H_{k+1})] - F(W_k, H_k) \leq -\alpha_k \nabla F_W(W_k, H_{k+1})^T \mathbb{E}_{\xi_k}[g(W_k, H_k; \xi_k)]$$

$$- \alpha_k \nabla F_H(W_k, H_k)^T \mathbb{E}_{\xi_k}[q(W_k, H_k; \xi_k)]$$

$$+ \frac{1}{2}\alpha_k^2 L\mathbb{E}_{\xi_k}[\|g(W_k, H_k; \xi_k)\|_2^2] + \frac{1}{2}\alpha_k^2 L\mathbb{E}_{\xi_k}[\|q(W_k, H_k; \xi_k)\|_2^2]$$

Combine Assumption 2 with Definition 4.6, we have the second moment of $g(W_k, H_k; \xi_k)$ and $q(W_k, H_k; \xi_k)$ satisfy

$$\mathbb{E}_{\xi_k}[\|g(W_k, H_k; \xi_k)\|_2^2] \leq M + M_G\|\nabla F_W(W_k, H_k)\|_2^2,$$

$$\mathbb{E}_{\xi_k}[\|q(W_k, H_k; \xi_k)\|_2^2] \leq M + M_G\|\nabla F_H(W_k, H_k)\|_2^2, \quad with \ M_G := M_V + \mu_G^2 \geq \mu^2 > 0.$$

Therefore we have

$$\mathbb{E}_{\xi_k}[F(W_{k+1}, H_{k+1})] - F(W_k, H_k) \leq -\mu\alpha_k [\|\nabla F_W(W_k, H_k)\|_2^2 + \|\nabla F_H(W_k, H_k)\|_2^2]$$

$$+ \frac{1}{2}\alpha_k^2 L[M + M_G(\|\nabla F_W(W_k, H_k)\|_2^2 + \|\nabla F_H(W_k, H_k)\|_2^2)]$$

$$\leq -(\mu - \frac{1}{2}\alpha_k L M_G)\alpha_k [\|\nabla F_W(W_k, H_k)\|_2^2 + \|\nabla F_H(W_k, H_k)\|_2^2]$$

$$+ \alpha_k^2 L M.$$

$\square$

**Proof of Theorem 1**

*Proof.* Taking the total expectation of Eq. A.1 and from the condition of $0 < \bar{\alpha} \le \frac{\mu}{LM_G}$,

$$\mathbb{E}[F(W_{k+1}, H_{k+1})] - \mathbb{E}[F(W_k, H_k)] \le -(\mu - \frac{1}{2}\bar{\alpha}LM_G)\bar{\alpha}\mathbb{E}\big[\|\nabla F(W_k, H_k)\|_2^2 + \|\nabla F_H(W_k, H_k)\|_2^2\big]$$
$$+ \bar{\alpha}^2 LM$$
$$\le -\frac{1}{2}\mu\bar{\alpha}\mathbb{E}\big[\|\nabla F_W(W_k, H_k)\|_2^2 + \|\nabla F_H(W_k, H_k)\|_2^2\big] + \bar{\alpha}^2 LM.$$

Summing both sides of this inequality for $k \in \{1, ..., K\}$ and recalling Assumption 2 (a) gives
$$F_{inf} - F(W_1, H_1) \le \mathbb{E}[F(W_{K+1}, H_{K+1})] - F(W_1)$$
$$\le -\frac{1}{2}\mu\bar{\alpha}\sum_{k=1}^{K}\mathbb{E}\big[\|\nabla F_W(W_k, H_k)\|_2^2 + \|\nabla F_H(W_k, H_k)\|_2^2\big] + K\bar{\alpha}^2 LM.$$

Rearranging above inequality and dividing further by $K$ yields the result. $\square$

**Proof of Theorem 2**

*Proof.* The second condition in Eq. 4.10 ensures that $\lim_{k\to\infty} \alpha_k = 0$, meaning that without loss of generality, we may assume that $\alpha_k LM_G \le \mu$ for all $k \in \mathbb{N}$. Taking the total expectation of Eq. A.1, we obtain

$$\mathbb{E}[F(W_{k+1}, H_{k+1})] - \mathbb{E}[F(W_k, H_k)] \le -(\mu - \frac{1}{2}\bar{\alpha}LM_G)\bar{\alpha}\mathbb{E}\big[\|\nabla F(W_k, H_k)\|_2^2 + \|\nabla F_H(W_k, H_k)\|_2^2\big]$$
$$\le -\frac{1}{2}\mu\alpha_k\mathbb{E}\big[\|\nabla F_W(W_k, H_k)\|_2^2 + \|\nabla F_H(W_k, H_k)\|_2^2\big] + \alpha_k^2 LM.$$

Summing both sides of this inequality for $k \in \{1, ..., K\}$ and recalling Assumption 2(a) gives
$$F_{inf} - F(W_1, H_1) \le \mathbb{E}[F(W_{K+1}, H_{K+1})] - F(W_1, H_1)$$
$$\le -\frac{1}{2}\mu\sum_{k=1}^{K}\alpha_k\mathbb{E}\big[\|\nabla F_W(W_k, H_k)\|_2^2 + \|\nabla F_H(W_k, H_k)\|_2^2\big] + LM\sum_{k=1}^{K}\alpha_k^2.$$

Rearranging the above inequality and dividing further by $\mu/2$, we obtain

$$\sum_{k=1}^{K}\alpha_k\mathbb{E}\big[\|\nabla F(W_k, H_k)\|_2^2 + \|\nabla F_H(W_k, H_k)\|_2^2\big] \le \frac{2(F(W_1, H_1) - F_{inf})}{\mu} + \frac{2LM}{\mu}\sum_{k=1}^{K}\alpha_k^2.$$

Then, we get the desired result by dividing $A_k$ on the both sides. $\square$

## B  Sufficient Direction Constant

Assumption 2(b) states that, in expectation, the vectors $g(W_k, H_k; \xi_k)$ and $q(W_k, H_k; \xi_k)$ are sufficient descent directions for $F$ from $W_k$ and $H_k$ with norms comparable to the norms of the gradients. It guarantees that the model moves towards the descending direction of the loss function. Following the experimental setup in Section 5.1, we demonstrate that the proposed method empirically satisfies Assumption 2(b), and visualize in Figure 7 the sufficient direction constant $\mu$ for the (partial) convolutional layers of the four models during the end-to-end training using TREC. For SqueezeNet and ResNet-34, we show one block as the representative, since the other blocks have similar performance.

Several insights can be drawn from Figure 7. (i) The value of $\mu$ of each convolutional layer is consistently greater than zero, indicating that Assumption 2(b) is satisfied, further ensuring the convergence of the TREC-equipped CNNs. (ii) The lower convolutional layers have smaller $\mu$ compared to the upper ones. (iii) The value of $\mu$ gradually increases through iterations. In fact, the

closer $\mu$ is to 1, the more the model moves toward the sufficient direction. Thus the gap between $\mu$ and 1 reflects the difference between the current descent direction of the model and the steepest descent direction [14]. The smaller values of $\mu$ in the lower convolutional layers in the early epochs indicate a larger difference in the descent direction. It is because the guidance obtained from the loss in the lower convolutinoal layers comes from back-propagation, which accumulates the disparities. Small $\mu$ values in the early epochs help the model avoid being trapped in local minimums, while large $\mu$ values in the later epochs help the model converge.

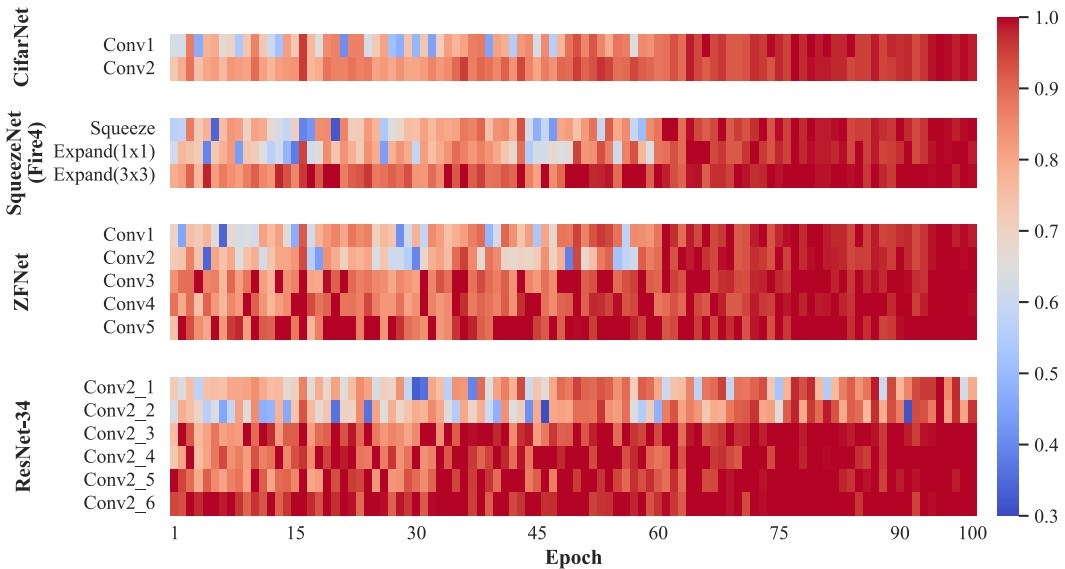

Figure 7: Sufficient direction constant $\mu$.

## C Single-Layer Performance

### C.1 Single-Layer Performance Benefits

Table 3: Single-layer performance benefits. $Conf.$ means configuration, $L$ is the width of sub-matrices, $H$ is the number of hash functions, $r_t$ represents the redundancy ratio, and $\Delta$ Acc is the accuracy difference.

| Network | Layer | Conf. L | Conf. H | $r_t$ | Speedup | ΔAcc. (vs. Conventional) | ΔAcc. (vs. Deep Reuse) |
|---|---|---|---|---|---|---|---|
| CifarNet | Conv1 | 5 | 15 | 0.9429 | 1.81× | -0.0132 | 0.0428 |
| | Conv2 | 10 | 10 | 0.7947 | 1.52× | -0.0077 | 0.0316 |
| Average | | | | 0.8688 | 1.66× | -0.0105 | 0.0372 |
| ZfNet | Conv1 | 147 | 5 | 0.9997 | 1.22× | -0.0011 | 0.0457 |
| | Conv2 | 300 | 5 | 0.9967 | 4.69× | -0.0037 | 0.0319 |
| | Conv3 | 432 | 5 | 0.9884 | 4.72× | -0.0076 | 0.0389 |
| | Conv4 | 432 | 5 | 0.9982 | 6.23× | -0.0105 | 0.0228 |
| | Conv5 | 288 | 5 | 0.9842 | 5.58× | -0.0068 | 0.0311 |
| Average | | | | 0.9934 | 4.49× | -0.0059 | 0.0341 |
| Vanilla SqueezeNet | Conv1 | 9 | 5 | 0.9969 | 1.27× | -0.0066 | 0.0311 |
| | Fire2 - squeeze | 96 | 4 | 0.9922 | 1.02× | -0.0179 | 0.0310 |
| | Fire2 - 1×1 expand | 8 | 5 | 0.9934 | 4.61× | 0.0006 | 0.0233 |
| | Fire2 - 3×3 expand | 48 | 5 | 0.9875 | 6.04× | 0.0040 | 0.0182 |
| | Fire3 - squeeze | 64 | 4 | 0.9877 | 1.30× | 0.0029 | 0.0138 |
| | Fire3 - 1×1 expand | 8 | 5 | 0.9932 | 4.51× | 0.0055 | 0.0132 |
| | Fire3 - 3×3 expand | 72 | 5 | 0.9876 | 6.43× | 0.0029 | 0.0138 |
| | Fire4 - squeeze | 64 | 4 | 0.9980 | 1.07× | -0.0062 | 0.0337 |
| | Fire4 - 1×1 expand | 16 | 5 | 0.9879 | 3.61× | 0.0037 | 0.0240 |

| Network | Layer | Conf. L | Conf. H | $r_t$ | Speedup | ΔAcc. (vs. Conventional) | ΔAcc. (vs. Deep Reuse) |
|---|---|---|---|---|---|---|---|
| | Fire4 - 3×3 expand | 144 | 5 | 0.9877 | 5.86× | 0.0035 | 0.0289 |
| | Fire5 - squeeze | 128 | 4 | 0.9844 | 1.05× | 0.0001 | 0.0139 |
| | Fire5 - 1×1 expand | 4 | 5 | 0.9906 | 4.96× | -0.0009 | 0.0151 |
| | Fire5 - 3×3 expand | 144 | 5 | 0.9500 | 3.25× | -0.0064 | 0.0162 |
| | Fire6 - squeeze | 32 | 5 | 0.9598 | 1.78× | -0.0008 | 0.1935 |
| | Fire6 - 1×1 expand | 6 | 5 | 0.9736 | 2.84× | 0.0064 | 0.0213 |
| | Fire6 - 3×3 expand | 54 | 5 | 0.9504 | 16.06× | -0.0078 | 0.0159 |
| | Fire7 - squeeze | 48 | 5 | 0.9600 | 1.39× | -0.0023 | 0.0146 |
| Vanilla | Fire7 - 1×1 expand | 6 | 5 | 0.9754 | 2.67× | 0.0049 | 0.0224 |
| SqueezeNet | Fire7 - 3×3 expand | 216 | 5 | 0.9523 | 16.95× | -0.0052 | 0.0105 |
| | Fire8 - squeeze | 8 | 5 | 0.9784 | 1.39× | -0.0063 | 0.0805 |
| | Fire8 - 1×1 expand | 4 | 5 | 0.9710 | 4.31× | 0.0065 | 0.0224 |
| | Fire8 - 3×3 expand | 288 | 5 | 0.9500 | 18.66× | -0.0042 | 0.0127 |
| | Fire9 - squeeze | 256 | 5 | 0.9250 | 1.07× | 0.0062 | 0.2365 |
| | Fire9 - 1×1 expand | 8 | 5 | 0.8563 | 1.10× | 0.0037 | 0.0201 |
| | Fire9 - 3×3 expand | 288 | 5 | 0.8156 | 12.63× | 0.0058 | 0.0148 |
| | Conv10 | 4 | 5 | 0.9235 | 1.32× | -0.0084 | 0.1901 |
| Average | | | | 0.9626 | 4.89× | -0.0006 | 0.0435 |
| | Conv1 | 9 | 5 | 0.9969 | 1.27× | 0.0122 | 0.0783 |
| | Fire2 - squeeze | 96 | 4 | 0.9926 | 1.07× | 0.0144 | 0.0559 |
| | Fire2 - 1×1 expand | 8 | 5 | 0.9914 | 1.41× | 0.0023 | 0.0149 |
| | Fire2 - 3×3 expand | 48 | 5 | 0.9897 | 5.90× | 0.0002 | 0.0156 |
| | Bypass1 - 1×1 | 32 | 5 | 0.9940 | 2.45× | 0.0071 | 0.0232 |
| | Fire3 - squeeze | 64 | 4 | 0.9877 | 1.07× | -0.0051 | 0.0145 |
| | Fire3 - 1×1 expand | 8 | 5 | 0.9918 | 2.30× | -0.0033 | 0.0167 |
| | Fire3 - 3×3 expand | 24 | 5 | 0.9876 | 5.85× | -0.0013 | 0.0222 |
| | Fire4 - squeeze | 64 | 5 | 0.9895 | 1.00× | -0.0040 | 0.0162 |
| | Fire4 - 1×1 expand | 16 | 5 | 0.9881 | 1.45× | -0.0037 | 0.0171 |
| | Fire4 - 3×3 expand | 48 | 5 | 0.9876 | 6.36× | 0.0012 | 0.0254 |
| | Bypass2 - 1×1 | 4 | 8 | 0.9933 | 2.53× | -0.0012 | 0.0277 |
| | Fire5 - squeeze | 64 | 4 | 0.9824 | 1.84× | -0.0036 | 0.0188 |
| | Fire5 - 1×1 expand | 16 | 5 | 0.9531 | 4.51× | 0.0016 | 0.0163 |
| SqueezeNet + | Fire5 - 3×3 expand | 48 | 5 | 0.9542 | 11.83× | 0.0001 | 0.0197 |
| Complex Bypass | Fire6 - squeeze | 128 | 5 | 0.9719 | 1.01× | -0.0018 | 0.0213 |
| | Fire6 - 1×1 expand | 32 | 5 | 0.9631 | 4.40× | -0.0071 | 0.0189 |
| | Fire6 - 3×3 expand | 72 | 5 | 0.9534 | 12.50× | 0.0002 | 0.0242 |
| | Bypass3 - 1×1 | 32 | 5 | 0.9768 | 5.29× | 0.0021 | 0.0192 |
| | Fire7 - squeeze | 48 | 5 | 0.9648 | 1.71× | 0.0003 | 0.0257 |
| | Fire7 - 1×1 expand | 24 | 5 | 0.9570 | 4.00× | -0.0046 | 0.0265 |
| | Fire7 - 3×3 expand | 48 | 5 | 0.9531 | 9.36× | -0.0025 | 0.0357 |
| | Fire8 - squeeze | 64 | 4 | 0.9685 | 1.03× | -0.0003 | 0.0299 |
| | Fire8 - 1×1 expand | 16 | 5 | 0.9645 | 2.15× | 0.0012 | 0.0164 |
| | Fire8 - 3×3 expand | 64 | 5 | 0.9550 | 7.57× | -0.0022 | 0.0188 |
| | Bypass4 - 1×1 | 128 | 5 | 0.9609 | 1.12× | -0.0045 | 0.0385 |
| | Fire9 - squeeze | 128 | 4 | 0.8938 | 1.01× | -0.0078 | 0.0138 |
| | Fire9 - 1×1 expand | 16 | 5 | 0.8391 | 1.53× | -0.0019 | 0.0188 |
| | Fire9 - 3×3 expand | 96 | 5 | 0.8406 | 11.64× | -0.0022 | 0.0235 |
| | Conv10 | 4 | 5 | 0.9438 | 1.29× | 0.0057 | 0.0326 |
| Average | | | | 0.9629 | 3.88× | -0.0003 | 0.0249 |
| | Conv1 | 25 | 5 | 0.8396 | 7.64× | -0.00063 | 0.0236 |
| | Conv2-1 | 144 | 5 | 0.9942 | 7.69× | -0.0010 | 0.0353 |
| | Conv2-2 | 144 | 5 | 0.9876 | 7.99× | -0.0002 | 0.0255 |
| ResNet | Conv2-3 | 144 | 5 | 0.9919 | 7.80× | -0.0027 | 0.0455 |
| (ImageNet-64×64) | Conv2-4 | 144 | 5 | 0.9880 | 7.98× | 0.0008 | 0.0301 |
| | Conv2-5 | 144 | 5 | 0.9884 | 7.96× | -0.0036 | 0.0201 |
| | Conv2-6 | 144 | 5 | 0.9876 | 7.99× | -0.0027 | 0.0475 |

| Network | Layer | Conf. L | H | $r_t$ | Speedup | ΔAcc. (vs. Conventional) | ΔAcc. (vs. Deep Reuse) |
|---|---|---|---|---|---|---|---|
| | Conv3-1 | 144 | 5 | 0.9510 | 6.75× | -0.0030 | 0.0250 |
| | Conv3-2 | 144 | 5 | 0.9579 | 7.18× | -0.0044 | 0.0169 |
| | Conv3-3 | 144 | 5 | 0.9500 | 6.87× | -0.0007 | 0.0302 |
| | Conv3-4 | 144 | 5 | 0.9537 | 7.02× | -0.0045 | 0.0190 |
| | Conv3-5 | 144 | 5 | 0.9528 | 6.98× | -0.0008 | 0.0398 |
| | Conv3-6 | 144 | 5 | 0.9554 | 7.09× | -0.0023 | 0.0229 |
| | Conv3-7 | 144 | 5 | 0.9557 | 7.10× | 0.0028 | 0.0262 |
| | Conv3-8 | 144 | 5 | 0.995 | 7.49× | -0.0011 | 0.0195 |
| | Conv4-1 | 288 | 5 | 0.9815 | 1.88× | -0.0003 | 0.0386 |
| | Conv4-2 | 72 | 5 | 0.9802 | 2.99× | -0.0036 | 0.0330 |
| | Conv4-3 | 72 | 5 | 0.9804 | 3.00× | -0.0022 | 0.0168 |
| | Conv4-4 | 72 | 5 | 0.9802 | 2.99× | -0.0002 | 0.0515 |
| ResNet | Conv4-5 | 72 | 5 | 0.9800 | 2.99× | -0.0015 | 0.0275 |
| (ImageNet-64×64) | Conv4-6 | 72 | 5 | 0.9802 | 2.99× | -0.0009 | 0.0607 |
| | Conv4-7 | 72 | 5 | 0.9812 | 3.01× | -0.0002 | 0.0311 |
| | Conv4-8 | 72 | 5 | 0.9800 | 2.99× | 0.0001 | 0.0376 |
| | Conv4-9 | 72 | 5 | 0.9803 | 3.00× | 0.0001 | 0.0196 |
| | Conv4-10 | 72 | 5 | 0.9804 | 3.00× | -0.0010 | 0.0427 |
| | Conv4-11 | 72 | 5 | 0.9838 | 3.06× | -0.0009 | 0.0500 |
| | Conv4-12 | 72 | 5 | 0.9800 | 2.99× | -0.0013 | 0.0271 |
| | Conv5-1 | 36 | 5 | 0.9279 | 1.54× | -0.0023 | 0.0506 |
| | Conv5-2 | 114 | 5 | 0.9239 | 1.20× | 0.0005 | 0.0327 |
| | Conv5-3 | 96 | 5 | 0.973 | 1.01× | -0.0011 | 0.0242 |
| | Conv5-4 | 96 | 5 | 0.925 | 1.01× | -0.0023 | 0.0441 |
| | Conv5-5 | 96 | 5 | 0.93 | 1.04× | -0.0007 | 0.0180 |
| | Conv5-6 | 96 | 5 | 0.9268 | 1.01× | -0.0011 | 0.0420 |
| Average | | | | 0.9630 | 4.64× | -0.0013 | 0.0326 |

## C.2 Sensitivity Study: Performance on Varying Batch Sizes

In this section, we perform an analysis of the impact of batch size on TREC inference performance. Due to the stringent memory limit of MCU, it is infeasible to run the inferences on larger batch sizes. We hence use servers for this experiment and focus on the amount of avoided redundancy. We visualize in Figure 8 the impact of batch size on the average redundancy ratio ($r_t$), the larger the more redundancy is avoided. It can be seen that as batch sizes increase, the redundancy ratios also increase. It is intuitive: A larger batch increases the number of neuron vectors in the input matrix, and hence increases the possibility for reusing the results. This suggests that TREC can bring in more computational savings at larger batch sizes.

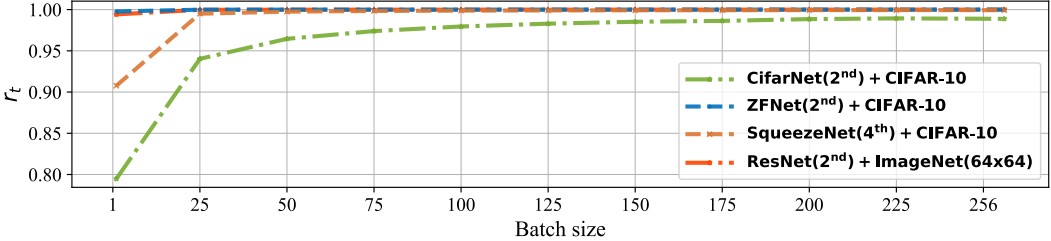

Figure 8: The impact of batch size on remaining ratio($r_t$).