# OpenReview forum: "TREC: Transient Redundancy Elimination-based Convolution"
_NeurIPS.cc/2022/Conference — NeurIPS 2022 Accept_

### Official Review · Reviewer_us1K · 2022-07-12

**Rating:** 5
**Confidence:** 3
**Soundness:** 3 good
**Presentation:** 3 good
**Contribution:** 3 good

**Summary:**

The paper proposes a convolution operation, dubbed TREC, with transient redundancy that is exploited during inference to reduce computation. Specifically, the authors use LSH to cluster layer inputs in the forward pass and use the cluster centers in the weight multiplication to compute the outputs. The authors propose a differentiable implementation for the LSH-based clustering which allows for training the TREC, thereby improving the accuracy compared to prior methods. In addition to empirical evaluations, the authors provide a theoretical analysis of the convergence and robustness of TREC.

**Questions:**

Please see the weaknesses above.

Questions and limitations:
- The reported accuracies for the Deep reuse paper are quite different from what is reported in the original paper (they claim no accuracy loss). Can the authors please clarify on the main differences in the evaluation setup that causes this difference?

-  Since the prior work Deep reuse, which is using a quite similar method, has evaluations on ImageNet, the authors are encouraged to add these evaluations to show that the scope of the proposed method is not limited to small datasets and the accuracy gains are persistent in more challenging benchmarks.

**Limitations:**

The authors have not discussed the limitations and potential negative social impact of their work

**Strengths And Weaknesses:**

Strengths:
- The paper is well written and the ideas are clearly explained.
- The combination of theoretical evidence with empirical results helps strengthen the claims.
- The ideas for making the clustering differentiable are interesting.

Weaknesses:
- The LSH-based clustering is previously proposed in the Deep Reuse paper. Therefore, the main contribution here is enabling backward propagation with a differentiable implementation for the clustering. The authors should explicitly clarify this in the paper and give proper reference to the prior art.
- The method is only evaluated on small-scale benchmarks (CIFAR-10) while prior work includes results on ImageNet. As such, a question rises regarding the applicability of the proposed method when evaluating larger scale datasets.

---

> ### Author Response · Authors · 2022-08-02
> **NeurIPS 2022 author response to reviewer us1K. Mainly focus on the motivation for TREC design; the experiment on large dataset; the accuracy of Deep Reuse.**
>
> # NeurIPS 2022 author response
>
> We thank the reviewers for insightful comments. We focus on answering the following concerns from reviewers:
> ## Motivation for TREC design
> As mentioned in the Introduction Section, in addition to designing a differentiable implementation of LSH clustering, TREC's contribution lies in proposing **a principled way to detect and avoid transient redundancy** (line 39). Deep Reuse is an ad-hoc approach to eliminating transient redundancy **during inferences** (line 27-30). It is an acceleration tool outside the DNN pipeline. In its use, data must be explicitly pulled out of the DNN pipeline, go through an LSH-based online clustering and matrix multiplication, and then get the results returned back to the DNN pipeline. On using LSH clustering, besides the large fluctuations in accuracy due to the randomly generated hashing vectors (line 30-31), Deep Reuse actually changes the inference process (line 37-38). Therefore, the weights of the original network may no longer best suit the inference.
>
> The difference brought about by TREC is that it incorporates the LSH clustering into the convolutional layer to construct a new convolution operator (line 39-41). This operator, through backpropagation-based training, learns the most effective hash vectors for eliminating redundancy and maintaining accuracy (line 47-51). **In this way, the weights of the network and the hashing vectors get updated hand in hand throughout the training, yielding the much better performance and stability for inference than Deep Reuse does.**
> ## Larger datasets
> Our experiment was on Microcontrollers while the previous Deep Reuse studies were on servers and laptops. Rather than gigabytes of memory, the Microcontrollers have only 2MB memory. The stringent resource on Microcontrollers stresses the importance of more effective redundancy removal, and also limits the size of the workload that can fit into it. Nevertheless, we managed to add an experiment of running the ***ResNet-34*** on ***ImageNet*** (64x64 since 224x224 and 256x256 run out of the MCU memory). By removing 96.3% of transient redundancy, TREC achieves an average single-layer speedup of 4.64x. When applied to the full ResNet, TREC achieves a speedup of 3.08x with 0.4% accuracy loss. The experimental results are presented in Section 5.3 and Appendix C, D.
> ## Accuracy of Deep Reuse
> From the authors of Deep Reuse, we learned that the results reported in their paper were the best they picked after many trials on the Cifar dataset, including an extensive tuning of the hyperparameters (e.g., learning rates) and trials of many randomly generated hash vectors. The first author of the Deep Reuse paper did the experiments several years ago and couldn't find the hyperparameters anymore; the results were not reproducible now. The results we included in the paper were the accuracy range we observed in 150 trials of Deep Reuse.
> ## Modifications to the paper
> 1. An experiment of running ResNet-34 on ImageNet (64x64). See Section 5 and Appendix C, D.
> 2. A verification of practical applications satisfying the sufficient direction assumption. See Appendix B.
> 3. An ablation analysis of the impact of batch size on single-layer performance. See Appendix C.2.

---

### Official Review · Reviewer_ubp7 · 2022-07-20

**Rating:** 5
**Confidence:** 3
**Soundness:** 2 fair
**Presentation:** 2 fair
**Contribution:** 3 good

**Summary:**

This paper proposes a principled way to detect and avoid transient redundancy for CNN inference. By introducing a new CNN operator named TREC, it fuses the detection and avoidance of transient redundancy into CNN, making them part of its inherent architecture. TREC comprises three components in series on top of Im2col+GEMM based convolution: LSH-based clustering that reduces the size of the input to the conversion, matrix multiplication of weights and the compressed input, and a recovery step that restores results to the original GEMM output sizes.

**Questions:**

- “Experiments show that by removing 96.22% transient redundancy, TREC achieves an average of 4.28x speedup compared to the conventional convolution operator.” What is the maximal possible speedup if 100% transient redundancy is removed?

- How would the BN layers (if the architecture has BN) be accommodated with such reduced input feature maps?

**Ethics Review Area:**

["I don’t know"]

**Limitations:**

No limitation is discussed.

**Strengths And Weaknesses:**

Strength
- By introducing a new CNN operator named TREC, it fuses the detection and avoidance of transient 40 redundancy into CNN, making them part of its inherent architecture
- TREC is compatible with both forward and backward propagations, enabling a plug-and-play replacement for convolutional layers in mainstream CNNs without additional effort.
- Theoretical analysis on the convergence is also provided.

Weakness
- The introduction of transient redundancy is not quite clear, it would be great to have some intuitive illustrations or numbers to demonstrate the redundancy. Why TREC makes the design and how it makes inference faster in sec 3.1 is also not quite clear.
- The evaluated CNN architecture seems not representative for common usage e.g., resnet.
- It is not clear whether architectures with BN layers can use TREC conv layers and how BN layers to be changed with the changed input feature maps.

---

> ### Author Response · Authors · 2022-08-02
> **NeurIPS 2022 author response to reviewer ubp7. Mainly focus on the motivation for TREC design; the definition of transient redundancy; the explanation of redundancy ratio; CNN choices; the accommodation to BN layers.**
>
> # NeurIPS 2022 author response
> We thank the reviewers for insightful comments. We focus on answering the following concerns from reviewers:
> ## Motivation for TREC design
> The motivation of TREC design is to optimize the inference performance on edge devices, and the LSH-based clustering it uses is inspired by Deep Reuse (line 39). Deep Reuse is an ad-hoc approach to eliminating transient redundancy during inferences (line 27-30). It is an acceleration tool outside the DNN pipeline. In its use, data must be explicitly pulled out of the DNN pipeline, go through an LSH-based online clustering and matrix multiplication, and then get the results returned back to the DNN pipeline. The LSH-based online clustering step is essential for effectively finding and avoiding redundancy in activation maps. It's a hashing vector based approach. As a stand-alone tool, Deep Reuse has a generic design, using randomly generated vectors as the hashing vectors in LSH. The randomness leads to large fluctuations in accuracy (more than 5%) across runs, meaning that the method may yield significantly different performance on one dataset in two different runs (line 30-31). It is detrimental to the reliability of ML models. Typically, nondeterministic methods are not preferred for real-world applications for their results are hard to reproduce and depend on. In addition, because Deep Reuse changes the inference process, the weights of the original network may no longer best suit the inference.
>
> The difference brought about by TREC is that it incorporates the LSH clustering into the convolutional layer to construct a new convolution operator (line 39-41). This operator, through backpropagation-based training, learns the most effective hash vectors for eliminating redundancy and maintaining accuracy (line 47-51). In this way, the weights of the network and the hashing vectors get updated hand in hand throughout the training, yielding the much better performance and stability for inference than Deep Reuse does. As the key difference between TREC and Deep Reuse is in the enabled backpropagation-based training of the LSH, the comparison presented in the paper between Deep Reuse and TREC is equivalent to an ablation analysis of the backpropagation step; our experimental results confirm its effectiveness.
> ## Definition of transient redundancy
> Transient redundancy arises from the activation maps and inputs which changes across inferences (line 21-24). It needs to be detected through online methods, and can be intuitively seen from the existence of similar patches within an image or activation map. Our results in the paper show that the redundancy can be as much as 96% (line 298).
> ## Explanation of “eliminate 96% of the transient redundancy”
> Sorry for not clarifying the definition of redundancy ratio. When we say that 96.22% of the transient redundancy is eliminated, we mean the LSH clustering has identified such an amount of similar vectors. It is expressed as the redundancy ratio ( $r_t$ ) in Section 4.3 to measure the size reduction of the input matrix (line 266-267). It is defined as $r_t=1-N_c/N$, where $N_c$ is the number of clusters and $N$ is the original number of rows (or neuron vectors) in the input matrix. Given such a definition, 100% transient ratio is unattainable, as it implies an empty centroid matrix. In addition, LSH, as an approximate clustering method, is inherently unable to achieve absolute accuracy and eliminate all redundancy.
> ## CNN choices
> Our experiment was on Microcontrollers while the previous Deep Reuse studies were on servers and laptops. Rather than gigabytes of memory, the Microcontrollers have only 2MB memory. The stringent resource on Microcontrollers stresses the importance of more effective redundancy removal, and also limits the size of the workload and network that can fit into it. Nevertheless, we managed to add an experiment of running the ResNet-34 on ImageNet (64x64 since 224x224 and 256x256 run out of the MCU memory). By removing 96.3% of transient redundancy, TREC achieves an average single-layer speedup of 4.64x. When applied to the full ResNet, TREC achieves a speedup of 3.08x with 0.4% accuracy loss. The results are presented in Section 5.3 and Appendix C, D.
> ## Accommodation to BN layers
> After TREC clusters the neuron vectors and computes the multiplication results of the reduced matrix (centroid matrix) and weight matrix, it uses the results to create a result matrix of the output dimensions of the original convolutional layer (line 81-82). By keeping the result shape unchanged, it ensures no changes needed on any subsequent network layers (including the BN layers).
> ## Modifications to the paper
> 1. An experiment of running ResNet-34 on ImageNet (64x64). See Section 5 and Appendix C, D.
> 2. A verification of practical applications satisfying the sufficient direction assumption. See Appendix B.
> 3. An ablation analysis of the impact of batch size on single-layer performance. See Appendix C.2.

---

### Official Review · Reviewer_PyXK · 2022-07-20

**Rating:** 6
**Confidence:** 4
**Soundness:** 3 good
**Presentation:** 2 fair
**Contribution:** 3 good

**Summary:**

This paper proposes, implements and tests a method to remove transient (data-point dependent) redundancy in the prpagation and backpropagation of convolutional neural networks.

**Questions:**

- Do we want to train on the edge device? Or do we want to fine-tune the model for the edge device such that it performs better after optimization than the original network?
- Any empirical evidence on data-sets besides CIFAR-10?
- Do the theoretical evidence give rise to concrete, quantitative bounds?

**Limitations:**

Yes the paper addresses the limitations of the methods. I can't see any particular negative societal impact specific to this line of work besides generic concerns that would apply to any methods that improve the efficiency of neural network evaluation in general.

**Strengths And Weaknesses:**

Originality: medium
This paper proposes the use LSH for removing (transient) data-point dependent computation cost of neural networks. While the idea has been used several times earlier (eg. in the Reformer [https://arxiv.org/abs/2001.04451]), its application for convolutional networks poses several technical challenges. While it is not too hard to see the solution to those challenges, it is still important and useful for giving a clear recipe for using them in practice and measuring their effectiveness in a real world scenario. In that sense, I think this paper makes a real contribution. Also the paper gives some theoretical guarantees in the limit which are new.

Quality: medium
This work proposes an LHS-based method to remove redundant computations in convolutional network. The method needs to do online clustering on the edge device, which results in a data-dependent reduction of computations. The paper reports an "acceleration" of the  layers applying this method by up to a 18X speedup. However this is not backed by a clear table of actual numbers, nor its dependence on batch size, neither ablation analysis or even mentioning the device on which this acceleration was reached. More thorough numbers are reported on the inference of the whole network for micro-controllers and the paper reports a total of 3.61X speedup on microcontrollers for the CIFAR-10X network, which would be impressive on its own and a good start.
However, the lack of evidence on data sets with larger than 32x32 images is disappointing and may question the relevance of the approach for real world use cases.
The theoretical analysis is nice and seems sound to me, but it works only in the limit and does not come with actual bounds, so its relevance for practical application seems questionable. Also it is unclear how far the theoretical bounds are from the real situation and whether the practical situations are within the range of the theoretical results being of any relevance whatsoever.

Clarity: medium
There are several unclear points in this paper:
- The motivation for implementing backpropagation at all is unclear: Do we want to train on the edge device? Or do we want to fine-tune the model for the edge device such that it performs better after optimization than the original network? I assume the latter, but this questions are left for the reader, while the paper puts a lot of emphasis on making backpropagation work. In the latter case (the network finetuned for higher quality evaluation), the paper lacks clear evidence and ablation analysis for the need/efficacy of that appraoch.

Significance: medium
If the method works it could give a clear recipe to improve the latency of CNN inference on certain compute-constrained edge devices. However, it is likely that with inroads of deep learning, specialized hardware will be used in most use cases, which might remove the need for such methods in the not too far future.

Generally, I think that the paper gives a valuable baseline and recipe for reducing the latency of inference of CNN on microcontrollers, but the experimental evidence given in this paper is not very conclusive, neither is the relevance of the theoretical evidence. I would raise my score in the presence of stronger practical evidence (evaluation on a dataset with larger images) and better motivation(/ablation analysis, if applies) for the backpropagation-related sections.

---

> ### Author Response · Authors · 2022-08-02
> **NeurIPS 2022 author response to reviewer PyXK. Mainly focus on the motivation for backpropagation; experiments on larger datasets; theoretical bounds.**
>
> # NeurIPS 2022 author response
> We thank the reviewers for insightful comments. We focus on answering the following concerns from reviewers:
> ## Motivation for backpropagation
> The motivation of TREC design is to optimize the inference performance of the model on edge devices, and the LSH-based clustering it uses is inspired by **Deep Reuse** (line 39). Deep Reuse is an ad-hoc approach to eliminating transient redundancy **during inferences** (line 27-30). It is an acceleration tool outside the DNN pipeline. In its use, data must be explicitly pulled out of the DNN pipeline, go through an LSH-based online clustering and matrix multiplication, and then get the results returned back to the DNN pipeline. The LSH-based online clustering step is essential for effectively finding and avoiding redundancy in activation maps. It's a hashing vector based approach. As a stand-alone tool, Deep Reuse has a generic design, using ***randomly generated vectors*** as the hashing vectors in LSH. The randomness leads to large fluctuations in accuracy (more than 5%) across runs, meaning that the method may yield significantly different performance on one dataset in two different runs (line 30-31). It is detrimental to the reliability of ML models. Typically, nondeterministic methods are not preferred for real-world applications for their results are hard to reproduce and depend on. In addition, because Deep Reuse changes the inference process, the weights of the original network may no longer best suit the inference.
>
> The difference brought about by TREC is that it incorporates the LSH clustering into the convolutional layer to construct a new convolution operator (line 39-41). **This operator, through backpropagation-based training, learns the most effective hash vectors for eliminating redundancy and maintaining accuracy (line 47-51).** In this way, the weights of the network and the hashing vectors get updated hand in hand throughout the training, yielding the much better performance and stability for inference than Deep Reuse does. **As the key difference between TREC and Deep Reuse is in the enabled backpropagation-based training of the LSH, the comparison presented in the paper between Deep Reuse and TREC is equivalent to an ablation analysis of the backpropagation step;** our experimental results confirm its effectiveness.
> ## Larger datasets
> Our experiment was on Microcontrollers while the previous Deep Reuse studies were on servers and laptops. Rather than gigabytes of memory, the Microcontrollers have only 2MB memory. The stringent resource on Microcontrollers stresses the importance of more effective redundancy removal, and also limits the size of the workload that can fit into it. Nevertheless, we managed to add an experiment of running the ResNet-34 on ImageNet (64x64 since 224x224 and 256x256 run out of the MCU memory). By removing 96.3% of transient redundancy, TREC achieves an average single-layer speedup of 4.64x. When applied to the full ResNet, TREC achieves a speedup of 3.08x with 0.4% accuracy loss. The experimental results are presented in Section 5.3 and Appendix C.
> ## Theoretical bounds
> According to Eq. 4.9 and 4.11 in the paper, as long as the practical application satisfies Assumptions 1 and 2, we can obtain a theoretical upper bound of the sum of squared gradients without considering the limit. The relevance of this theoretical upper bound to practical applications is reflected in the values of the constants such as $L$, $\mu$, $M$, and the specific setting of the learning rate. From Eqs. 4.5-4.8, the constants $L$, $M$, and $M_G = M_v + \mu_G^2$, which are greater than 0, exist according to the L2-norm. And through experimental verification, we find that the constant $\mu$ is always greater than 0 during the training process and gradually converges to 1 with time, satisfying the sufficient direction assumption. We add a figure in Appendix B to show the trend of the constant $\mu$.
> ## Answers to minor issues
> ### Detailed single-layer performance
> Both single-layer and end-to-end performance are averages obtained by inference with a batch size of 1 on the MCU, since large batch sizes can exhaust MCU memory. We will clarify the setting in Section 5. Also, we will illustrate the impact of batch size on performance in Appendix C.2 by measuring the redundancy ratio on the server. Actual numbers for single layer performance see link: https://anonymous.4open.science/r/TREC_Single_Layer-AC65.
> ### The need for edge inference acceleration
> With the rise of cloud computing and the AIoT, microcontrollers have dominated the market due to their low power consumption. It is estimated that 250 billion microcontrollers are already in use, and their volumes are expected to reach 1 trillion in various industries by 2035. Therefore, the computing demand of MCUs will last for a long period of time. Furthermore, since TREC is not hardware dependent, we consider applying TREC to other specialized hardware as a future study.

---

### Official Review · Reviewer_wscN · 2022-07-25

**Rating:** 5
**Confidence:** 3
**Ethics Flag:** Yes
**Soundness:** 4 excellent
**Presentation:** 4 excellent
**Contribution:** 2 fair

**Summary:**

The paper proposes a method to reduce the number of operations in a CNN by reducing the input to subsequent layers via the clustering of (intermediate) input features/pixels. The authors claim to incorporate the method to be more inherently combined with network architecture coining it, transient redundancy convolution. In particular, the method clusters the rows (convolutions) and replaces the original input with clustered centroids. The paper addresses the issues of how to perform backpropagation with the proposed formulation and provides guarantees for robustness and convergence.


**Questions:**

In my understanding by clustering, we reduce the number of convolution operations (or scalar products given the GEMM). Theoretically, it sounds promising but in practice, can you actually obtain any speed-ups? Do you also decrease the number of channels of your model (maybe if you remove all the convolutions for a given channel)? Are you planning to release the code?

I'm not sure where the terms come from. For example, in pruning literature, I never heard of the term “lasting redundancy”. Also, the phrases “eliminate 96% of the transient redundancy” can be puzzling without the proper definition in terms of FLOPs, for example.  I'd be cautious with using new terms, which may create confusion.


**Limitations:**

Little is said about the work’s limitations.


**Strengths And Weaknesses:**

The idea of clustering the inputs on the fly is interesting compared to the reducing the size of the input to NN only, or compressing the network and fixing its parameters. In TREC for every input data, we perform individual clustering at every stage of the forward pass to reduce further convolution operations. As a result, the redundancy is computed at every stage of the inference which makes it more flexible than reducing the size of the input to NN or decreasing the size of the network.

The analysis of complexity, convergence, and robustness is giving better insight into the theoretical guarantees of the algorithm.

The paper seems to heavily rely on the DeepReuse. It's following the same ideas, for example, using the same method for “clustering”, LSH hashing, same parameters L and H. Please state clearly the differences between the two approaches, and what elements of your method cause the improvements in TREC as compared to DeepReuse (also, I hardly see the randomness as the big disadvantage of DeepReuse).

---

> ### Author Response · Authors · 2022-08-02
> **NeurIPS 2022 author response to reviewer wscN. Mainly focus on the differences between TREC and Deep Reuse; the definition of lasting and transient redundancy; the explanation of redundancy ratio.**
>
> # NeurIPS 2022 author response
>
> We thank the reviewers for insightful comments. We focus on answering the following concerns from reviewers:
> ## Difference between TREC and Deep Reuse
> Deep Reuse is an ad-hoc approach to eliminating transient redundancy during inferences (*line 27-31*). It is an acceleration tool outside the DNN pipeline. In its use, data must be explicitly pulled out of the DNN pipeline, go through an LSH-based online clustering and matrix multiplication, and then get the results returned back to the DNN pipeline. The LSH-based online clustering step is essential for effectively finding and avoiding redundancy in activation maps. It's a hashing vector based approach. As a stand-alone tool, Deep Reuse has **a generic design**, using ***randomly generated vectors*** as the hashing vectors in LSH. The randomness leads to large fluctuations in accuracy (more than 5%) across runs, meaning that the method may yield significantly different performance on one dataset in two different runs. It is detrimental to the reliability of ML models. Typically, nondeterministic methods are not preferred for real-world applications for their results are hard to reproduce and depend on. In addition, because Deep Reuse changes the inference process, the weights of the original network may no longer best suit the inference.
>
> The difference brought about by TREC is that it incorporates the LSH clustering into the convolutional layer to construct a new convolution operator (*line 39-41*). This operator, through **backpropagation-based training**, learns ***the most effective hash vectors*** for eliminating redundancy and maintaining accuracy (*line 47-51*). In this way, the weights of the network and the hashing vectors get updated hand in hand throughout the training, yielding the much better performance and stability for inference than Deep Reuse does. As the key difference between TREC and Deep Reuse is in the enabled backpropagation-based training of the LSH, the comparison presented in the paper between Deep Reuse and TREC is equivalent to an ablation analysis of the backpropagation step; our experimental results confirm its effectiveness.
>
> ## Definition of lasting redundancy and transient redundancy
> We coined the two terms to stress the differences of the two kinds of redundancies in DNN, which are essential for understanding the needs for different approaches. We believe this classification is intuitive. Lasting redundancy arises from model parameters which usually stay stable across inferences (*line 18-21*), while transient redundancy arises from the activation maps and inputs which changes across inferences (*line 21-24*). Because of the different natures, redundancy in the former can be detected through offline compression methods, but the latter needs to be detected through online methods during inferences. Transient redundancy can be intuitively seen from the existence of similar patches within an image or activation map. Our results in the paper show that the redundancy can be as much as 96% (*line 298*).
> ## Explanation of “eliminate 96% of the transient redundancy”
> When we say that 96.22% of the transient redundancy is eliminated, we mean the LSH clustering has identified such an amount of similar vectors. It is expressed as the redundancy ratio ( $r_t$ ) in Section 4.3 to measure the size reduction of the input matrix (*line 266-267*). It is defined as $r_t=1-N_c/N$, where $N_c$ is the number of clusters and $N$ is the original number of rows (or neuron vectors) in the input matrix. If the reduction of transient redundancy is to be defined in terms of ***FLOPs***, given that the dimensions of the input and weight matrices unfolded by Im2col are $N \times K$ and $K \times M$, the redundant computation reduced by the matmul step is $r_t \cdot N \cdot K \cdot M$.
>
> ## Answers to minor issues
> ### Actual speedups
> With the sublinear performance of LSH, TREC achieves an average speedup of 4.4x compared to conventional convolution operator (*Section 5.2 and Appendix C*). When applied to full networks, TREC achieves an average speedup of 3.51x (*Section 5.3*).
> ### Channel pruning
> Channel pruning is used only on ZfNet so that it can fit into the Microcontroller (*line 291*). In our speedup comparisons, the baseline and our TREC performance are on the same pruned ZfNet; in another word, the speedups come from the more effective redundancy removal by TREC, not channel pruning.
> ### Code Release
> Yes, we plan to release the code along with the models and datasets.
> ## Modifications to the paper
> 1. An experiment of running ResNet-34 on ImageNet (64x64, since 224x224 and 256x256 run out of the MCU memory). See *Section 5* and *Appendix C, D*.
> 2. A verification of practical applications satisfying the sufficient direction assumption. See *Appendix B*.
> 3. An ablation analysis of the impact of batch size on single-layer performance. See *Appendix C.2*.

---

### Meta-Review · Area_Chair_MLw2 · 2022-08-27

**Recommendation:** Accept
**Confidence:** Less certain

**Metareview:**

This work proposes a redundancy pruning mechanism for convolutional neural networks to optimize their performance for extremely resource-constrained edge devices, such as microcontrollers.

The work is based on a novel gradient-optimized locality sensitive hashing approach that removes a lot of the indeterminacy of previous approaches. This method can achieve reliable high performance while improving the network inference latency significantly (eg 4x in some circumstance)

I think that the alone the idea of tuning LSH via SGD directly for clustering purposes is a good one and is of interest to the wider community.

**Award:**

No

---

### Decision · Program_Chairs · 2022-09-14

Accept